# Unconventional Treatments for Pancreatic Cancer: A Systematic Review

**DOI:** 10.3390/cancers17091437

**Published:** 2025-04-25

**Authors:** Roberta Vella, Alessandro Giardino, Erica Pizzocaro, Isabella Frigerio, Elisa Bannone, Salvatore Vieni, Giovanni Butturini

**Affiliations:** 1HPB Surgery, Pederzoli Hospital, Peschiera del Garda, 37019 Verona, Italy; roberta.vella02@unipa.it (R.V.); alessandro.giardino@ospedalepederzoli.it (A.G.); isabella.frigerio@ospedalepederzoli.it (I.F.); elisa.bannone@ospedalepederzoli.it (E.B.); giovanni.butturini@ospedalepederzoli.it (G.B.); 2Department of Precision Medicine in Medical, Surgical, and Critical Care, (Me.Pre.C.C.), University of Palermo, 90133 Palermo, Italy; salvatore.vieni@unipa.it; 3PhD School of Applied Medical-Surgical Sciences, University Tor Vergata, 00133 Rome, Italy; 4Collegium Medicum, SAN University, 90-012 Lodz, Poland

**Keywords:** pancreatic cancer, systematic review, complementary medicine, unconventional treatment, precision oncology

## Abstract

The use of complementary and alternative medicine (CAM) is increasingly common among pancreatic cancer patients due to cultural influences, psychological factors, and the poor prognosis of the disease. Many patients seek unconventional treatments to manage symptoms, improve quality of life, and enhance adherence to standard therapies. However, there is uncertainty regarding the effectiveness and safety of unconventional treatments due to a lack of scientific validation. This systematic review examines various CAM therapies, including curcumin, mistletoe extract, Chinese herbal medicine, and electroacupuncture, highlighting their potential benefits as complementary treatments within integrative oncology. Despite some promising findings, the absence of rigorous clinical trials hinders their integration into standard cancer care. This review underscores the widespread use of CAM and stresses the need for further scientific research. Proper regulation, adherence to ethical principles, and high-quality studies are essential to ensure these treatments can be safely incorporated into holistic patient care.

## 1. Introduction

Unconventional treatments encompass a broad spectrum of medical practices that operate outside the methods of conventional medicine [1]. They include complementary and alternative medicine (CAM) approaches, such as the use of natural products, dietary supplements, probiotics, whole medical systems [2], lifestyle interventions, body-based practices, and physical therapies.

Unconventional treatments have gained increasing popularity, even among cancer patients. Studies report that up to 70.2% of patients undergoing chemotherapy incorporate unconventional treatments [3] into their treatment schedule to manage symptoms and improve adherence to conventional therapy.

Pancreatic cancer is a pressing public health issue, ranking among the leading causes of cancer-related mortality, with a dismal 5-year survival rate of 11.5% [3,4]. Around 80% of pancreatic cancer patients are diagnosed at an advanced stage [5], precluding surgical resection and limiting treatment options to systemic therapies that continue to demonstrate suboptimal long-term efficacy. Beyond its poor prognosis, pancreatic cancer is often associated with debilitating symptoms, such as pain and diarrhea [6], prompting many patients to seek relief through CAM.

The widespread availability and aggressive promotion of unconventional treatments have further contributed to their growing popularity and use. Nevertheless, the efficacy and safety of these interventions are unknown and require rigorous investigation.

This systematic review aims to evaluate the effects of unconventional treatments in pancreatic cancer patients, either as standalone interventions or when used in combination with standard therapies.

The findings may provide valuable insights into the role of unconventional treatments within an integrative approach to managing pancreatic cancer while also investigating their efficacy and potential risks.

## 2. Materials and Methods

### 2.1. Information Sources and Search Strategy

An electronic, systematic, and comprehensive literature review was conducted and reported following the PRISMA 2020 and AMSTAR 2 (assessing the methodological quality of systematic reviews) guidelines [7,8]. The search covered all relevant research literature published from 2010 through March 2024 using MEDLINE (PubMed), Scopus, Google Scholar, and Embase. References from the included studies were also checked to identify any additional relevant papers. The following search terms were used: “Unconventional treatment”, “Complementary medicine”, “Alternative medicine”, “Pancreatic Cancer”. The full search strategy for PubMed is detailed in Appendix B. The study protocol was registered on PROSPERO (ID: CRD42024600460).

### 2.2. Eligibility Criteria

Both published and ongoing randomized controlled trials (RCTs) and prospective and retrospective studies evaluating the effects of unconventional treatments for pancreatic cancer were included. Interventional studies had to assess the impact of an unconventional treatment modality, either as a standalone intervention or used in combination with standard therapies. Reports not in the English language, case reports, and studies not involving humans were excluded.

### 2.3. Selection and Data Collection Processes

The screening process was performed using the free online app Rayyan (http://rayyan.qcri.org, accessed on 12 October 2024). After removing duplicates, two authors (RV and EP) independently screened titles and abstracts to identify potentially eligible studies. The full texts of the selected studies were retrieved and independently assessed for eligibility by the same authors. Any uncertainties regarding the inclusion of the studies were resolved by consensus and, if necessary, by consulting a third author (GB).

### 2.4. Data Items

Data were extracted into an Excel sheet (Microsoft Excel Version 17, Microsoft Corporation 19) and analyzed using RevMan, version 5.4. Extracted variables included author, year of publication, study design, study location, trial registration number, funding or sponsorship, blinding, number of included patients, study aim, inclusion and exclusion criteria, and details of the unconventional treatment. The demographic data (i.e., age, gender) of patients from the included studies were reported as mean (with standard deviation) or median (with interquartile range) for continuous variables and as numbers with percentages for categorical variables.

### 2.5. Outcome Measures

#### Primary Outcomes

The following outcome measures were retrieved if reported by each study:Overall survival;Quality of life;Symptom relief.

For these variables, the frequency of the effect was extracted, and when available, the effect size was reported as a risk ratio (RR) with a 95% confidence interval (CI).

The secondary outcome was as follows:Tolerability of the unconventional treatment, defined as the incidence of side effects.

### 2.6. Study Risk of Bias Assessment

The risk of bias was assessed using the Cochrane Risk of Bias Assessment Tool 2 (Cochrane collaboration, London, UK, 2019) for RCTs and tabulated using the ROBVIS tool. The assessment considered five domains: sequence generation, allocation concealment, blinding, incomplete outcome data, selective outcome reporting, and other potential sources of systematic bias. For each study, the risk of bias was ranked as low, high, or of some concern.

Due to the heterogeneity in both the effect size and type of unconventional treatment (e.g., differences in dose and frequency of intervention, application methods, concomitant conventional treatments, and patients’ characteristics), a meta-analysis was not conducted. Therefore, a systematic narrative synthesis of the study results is provided.

## 3. Results

A total of 966 records were retrieved, including 7 identified through snowball searching. After de-duplication, 962 records underwent title and abstract screening, resulting in 55 studies selected for full-text review. Of these, 21 studies met the eligibility criteria and were included in the systematic review, while 33 were excluded (Appendix A, table of excluded studies). The flow diagram of the study selection process is presented in Figure 1.

### 3.1. Characteristics of Included Studies

The 21 included studies provided data from 3095 patients, with most studies conducted in China between 2010 and 2024 [9,10,11,12,13,14,15,16,17,18,19]. The general characteristics of the included studies are summarized in Table 1.

The included studies evaluated the use of various unconventional treatments in pancreatic cancer patients:Chinese herbal medicine (CHM) [9,10,11,12,13,14,15,20,21,22];Mistletoe [23,24,25,26,27];Curcumin [16,17,28,29];Unconventional therapies for pain management in pancreatic cancer [18,19].

Demographic data of patient cohorts from the included studies are displayed in Table 2.

### 3.2. Quality of Included Studies

Among the included studies, five were RCTs [10,18,19,25,26], while ten were retrospective cohort studies [9,11,12,13,14,21,22,23,24,27] and six were prospective cohort studies [15,16,17,20,28,29].

The overall quality of the RCTs was judged to have some concerns (Figure 2).

Four RCTs [18,22,25,26] had a low risk of bias, while one had some concerns [19] due to inadequate allocation and an absence of participant blinding, which may have biased the results (Figure 2). Trial protocols were available only for three RCTs [22,25,26]. One RCT [22] received funding from a company involved in producing the tested CHM (Appendix A).

### 3.3. Ongoing Studies

An ongoing double-blind randomized phase II trial [30] is investigating active hexose correlated compound (AHCC), a standardized extract of cultured Lentinula edodes mycelia, amino acids, minerals, lipids, and polysaccharides, in patients with borderline resectable or resectable pancreatic cancer undergoing neoadjuvant therapy (scheme: gemcitabine plus S-1). The study aims to determine whether AHCC improves 2-year disease-free survival (DFS) compared to placebo and is expected to clarify the role of AHCC in mitigating chemotherapy-related adverse events and improving survival.

### 3.4. Results by Treatment Modality

#### 3.4.1. Chinese Herbal Medicine

Ten studies evaluated the effects of CHM [9,10,11,12,13,14,15,20,21,22] in patients with advanced and metastatic pancreatic cancer. Most studies reported that CHM improves survival and quality of life [9,11,12,13,14,15,20,21,22,31], potentially through synergistic effects with conventional therapies [12]. However, the treatment regimens used varied, with some studies implementing long-term continuous administration [9] while others employed cyclic administrations over shorter durations [20]. CHM was assessed both as a standalone therapy [9] and in combination with conventional gemcitabine-based chemotherapy [13,21]. The reported median survival times varied based on the study population, with 4.7 months observed in patients with metastatic liver disease [14] and 15.2 months in a broader cohort including both resectable and metastatic disease [9] (Table 3).

#### 3.4.2. Mistletoe

Five studies [23,24,25,26,27], including two RCTs [25,26], assessed the effect of mistletoe (i.e., *Viscum album* L.) in advanced pancreatic cancer, evaluating overall survival [23,24,25,26,27], quality of life [25,26], symptom relief [25,26], safety, and cost-effectiveness [24]. The results were mixed. All studies were conducted in Europe and included patients with locally advanced or metastatic disease. Different types of mistletoe extract (ME) were used and included Helixor [23,24,27], Abnoba viscum [23,24,27], or Iscador [23,24,25,26] mistletoe, with subcutaneous [23,25,26,27], intravenous [23,27], or intratumoral [23,27] administration. MEs were used as standalone treatments [25,27,32] or alongside surgery [23] or radiotherapy [23,27]. The dose of ME varied among patients and was progressively increased to each participant’s level of tolerability.

The RCT reported by Troger et al. [25], the multicenter observational study conducted by Axtner et al. [27], and the retrospective cohort study from Thronicke et al. [24] all demonstrated a survival benefit when patients were treated with ME. OS was longer when chemotherapy was combined with ME than when used alone (12.1 months vs. 7.3 months, *p* value not reported [27]; 8.43 months vs. 5.63 months, *p* value not reported [24]). Among patients who did not receive chemotherapy, ME improved OS from 2.7 to 4.8 months (*p* < 0.001) [33] in an RCT and from 2.5 to 5.4 months (*p* value not reported) [27] in an observational study compared to best supportive care (Table 3). Schad et al. [23] reported median survival times of 11.8 and 8.3 months for stage III and IV pancreatic cancer, respectively, in patients treated with chemotherapy and intratumoral ME injections; however, the absence of a control group limited the strength of the conclusions from these data (Table 3).

ME improved the quality of life of patients with advanced pancreatic cancer in one trial [25], enhancing global health status and social, cognitive, and physical functioning, while reducing symptoms such as pain, fatigue, appetite loss, insomnia, nausea, and vomiting, with no reported side effects. In contrast, the single RCT reported by Wode et al. [26] found no survival or quality-of-life benefits, reporting a median OS of 7.8 months in the ME group vs. 8.3 months in the standard care placebo group (*p* = 0.88). The number, severity, and outcomes of adverse events were comparable [23,25,27], except for a greater incidence of local skin reactions at ME injection sites (66% vs. 1%, *p* value not reported [26]).

A cost analysis [24] suggested ME prolonged survival compared to standard chemotherapy alone, leading to increased hospitalizations. However, when adjusted for mean OS, the average hospital costs were lower for the combined ME and chemotherapy treatment compared to chemotherapy alone.

#### 3.4.3. Curcumin

Four single-cohort studies [16,17,28,29] investigated curcumin or curcumin derivatives as a complementary therapy in combination with gemcitabine for pancreatic cancer. One study [17] examined Theracurmin, a highly bioavailable curcumin derivative, for its safety and pharmacokinetics, another [28] used a curcumin phytosome complex to enhance gemcitabine efficacy, and two studies [16,29] used a mixture of curcuminoids containing curcumin and curcumin derivates (demethoxycurcumin and bisdemethoxycurcumin). The recommended daily dose of curcumin (8 g/day) was used in all studies [16,17,28], but the study from Epelbaum et al. [29] reported gastrointestinal toxicity requiring a dose reduction to 4 g/day and limiting the treatment’s feasibility.

All included studies [16,17,28,29] reported some benefits from adding curcumin to gemcitabine-based chemotherapy in advanced pancreatic cancer. In patients with disease progression after chemotherapy and no alternative treatment options, two studies [16,17] found no partial or complete responses with curcumin and chemotherapy. However, in patients with locally advanced or metastatic disease, two studies reported response rates of 27.3% [28] and 9.1% [29] when curcumin was combined with gemcitabine; a median OS of 10.2 months was found in one of these studies [28]. All studies reported a subset of patients achieving stable disease, the rates of which were 28% [16], 25% [17], 34% [28], and 36.4% [29].

Additionally, one study [17] reported an improvement in fatigue- and function-related quality-of-life scores when curcumin was added to the chemotherapy regimen.

#### 3.4.4. Unconventional Therapies for Pain Management in Pancreatic Cancer

Two RCTs evaluated the efficacy of electroacupuncture in managing pain in patients with advanced pancreatic cancer [18,19]. In both studies, electroacupuncture was administered over three consecutive days, demonstrating [18,19] a significant reduction in pain intensity compared to the control group (*p* < 0.001 in both studies, Table 3).

In both trials [18,19], patients in the intervention group received electroacupuncture with analgesic medications, including opioids. One study [18] used a placebo control, while the other [19] compared electroacupuncture with standard analgesic treatment alone. The treatment parameters, including needle insertion points, current frequency and intensity, and treatment duration, were comparable across both studies [18,19].

No adverse events or infections were reported at the treated sites. Pain severity was assessed using the numeric rating scale (NRS) at one day post-treatment in one study [19] and at two days post-treatment in the other (18). Notably, the trial conducted by Chen et al. [18] included patients with mild to moderate pain, whereas Tian et al. [19] also enrolled patients with severe pain (NRS 7–8).

## 4. Discussion

The growing interest in unconventional treatments for pancreatic cancer is driven by a complex interplay of cultural influences, psychological factors, and the challenges associated with the disease’s poor prognosis [31,32]. With an overall five-year survival rate of approximately 6% [33,34], many patients with advanced disease explore alternative or concomitant options with the aim of improving the outcomes of conventional therapies. The challenging prognosis of pancreatic cancer may lead some patients to distrust conventional therapies and seek unverified alternative treatments. As a counter to this, integrative oncology offers a pathway to bridge this gap, fostering trust through transparent and empathetic communication, rigorous research, and ethical practices. By addressing patients’ concerns and vulnerabilities beyond symptom relief, this comprehensive, patient-centered approach aims to achieve the best outcomes, improving both survival and quality of life.

This systematic review emphasizes the increasingly prominent role of unconventional treatments within integrative oncology for advanced pancreatic cancer. Complementary and alternative medicine, particularly Chinese herbal medicine and natural compounds, such as mistletoe and curcumin, has shown potential for providing symptomatic relief and aiding in disease control (Figure 3).

However, heterogeneity in dosage, routes of administration, treatment modalities, and effect size hamper the robustness and generalizability of these findings.

### 4.1. Chinese Herbal Medicine

CHM has been long utilized for its immunomodulatory properties [35], including the ability of some CHMs to enhance immune responses by acting as immune checkpoint inhibitors, modulating the tumor microenvironment, and reducing chemotherapy resistance [36,37,38]. While traditionally rooted in Eastern medicine, its adoption in Western integrative oncology is gaining traction, particularly for mitigating chemotherapy-induced side effects and enhancing patient tolerance for standard treatments.

The studies included in this review [9,10,11,12,13,14,15,20,21,22] consistently indicated that CHM, when used as an adjunct to conventional therapies, improved median survival and one-year survival rates. Additionally, CHM appeared to enhance treatment tolerability by reducing chemotherapy-associated side effects [9]. Despite these promising results, variations in CHM formulations and study methodologies underscore the need for further high-quality research to confirm the efficacy of CHMs and establish optimal treatment protocols.

### 4.2. Mistletoe Extract

Mistletoe extract (*Viscum album* L.) contains various bioactive compounds, including lectins, viscotoxins, polysaccharides, and flavonoids [39], and has purported cytotoxic, immunostimulatory, and anti-angiogenic properties [40,41,42]. The use of ME in cancer treatment traces back to the early 20th century, when it was introduced by Rudolf Steiner in the pseudoscientific context of anthroposophical medicine [32]. Still used in European integrative oncology [43], its potential role in pancreatic cancer remains unclear due to conflicting and substandard study results. Some research suggests benefits in terms of quality of life, symptom management, and survival when ME is used either alone [25] or alongside chemotherapy [23,24,27], while another study [26] reported no significant impact, particularly in palliative care settings. Differences in healthcare systems and levels of palliative care integration may have contributed to these discrepancies. Additionally, concerns regarding study design, including open-label methodologies [25] and potential study population overlaps [23,24,27], further complicate the interpretation of findings, raising questions about the independence of datasets and the robustness of outcomes.

Despite some promising yet exploratory results in terms of OS from the study of Troger et al. [25], which challenge the current treatment options for patients with pancreatic cancer who are not candidates for curative therapies, further rigorously designed and controlled trials are essential to clarify the clinical value of ME in multimodal pancreatic cancer treatments. Future studies should focus on optimizing administration routes, identifying the most effective MEs for improving outcomes, and refining dosing strategies to minimize adverse events while maximizing therapeutic efficacy.

### 4.3. Curcumin

Curcumin, the most biologically active constituent of Curcuma longa, is widely used as a spice in Asian countries and is a key component of Ayurvedic medicine [44]. It exhibits several pharmacological effects, including antioxidant, anti-inflammatory, and chemopreventive properties, through the modulation of multiple signaling pathways [45,46,47]. Preliminary clinical trials have suggested that curcumin may slow tumor progression in pancreatic cancer patients when used in combination with gemcitabine [48,49]. The studies included in this review [16,17,28,29] support these preclinical findings, demonstrating a potential survival benefit when curcumin is combined with gemcitabine; the median survival in patients treated with this combination exceeds the historical median survival of 5.71 months [50] observed when gemcitabine is used as a monotherapy in patients with advanced pancreatic cancer. This outcome is comparable to those achieved with more advanced gemcitabine-based chemotherapy regimens [51], such as nanoparticle albumin-bound paclitaxel plus gemcitabine (nab-P + G). However, the small sample sizes and absence of control groups in these studies mean caution is warranted when interpreting these findings. Further large-scale trials are necessary to confirm the efficacy and safety of the curcumin–gemcitabine combination, optimize formulations, and establish standardized dosage regimens to maximize therapeutic benefit.

### 4.4. Unconventional Therapies for Pain Management in Pancreatic Cancer

Pain affects approximately 75% of patients at diagnosis [52] and stems from several factors, including tissue damage, neurogenic inflammation, ductal obstruction, and tumor infiltration [53,54]. Pain management is challenging due to the age profile of the patients, aggressive polychemotherapy, and the neurotropism of the disease [55]. Standard pain management relies on opioid therapy, with additional treatments such as radiotherapy and celiac plexus neurolysis [56] offering potential analgesic benefits. These conventional approaches, however, are not uniformly effective and are often associated with undesirable side effects. As a result, complementary and alternative methods have been investigated [57]. Acupuncture, a traditional Chinese medicine therapy recognized by the World Health Organization (WHO) and practiced in over 78 countries [58], involves the insertion of needles at predefined anatomical points to achieve therapeutic effects [59]. Electroacupuncture or transcutaneous electrical acupoint stimulation (TEAS), a variant that combines acupuncture with transcutaneous nerve electrical nerve stimulation, delivers low-frequency pulse currents at specific points (i.e., Jiaji points), targeting multiple pain mechanisms involving the peripheral, spinal, and supraspinal pathways [60].

Initial evidence supports the efficacy of acupuncture as an adjunctive analgesic for cancer pain management [61,62]. However, heterogeneity in acupuncture protocols, such as variations in style, needling techniques, number of sessions, and duration, hamper the generalizability of findings. Two RCTs included in this review [18,19] demonstrate that electroacupuncture significantly reduced pain intensity in patients with advanced pancreatic cancer. While these RCTs did not provide a direct comparison between electroacupuncture and opioid pharmacotherapy—the current standard of care for pancreatic cancer pain—the findings [18,19] do highlight the potential of electroacupuncture to alleviate cancer pain without adverse effects. Both RCTs [18,19] employ the same anatomical acupuncture points, enhancing the potential for consistent clinical application. In light of these findings, electroacupuncture may be a valuable complementary or alternative treatment to opioids, offering effective pain relief with fewer side effects [63]. Furthermore, electroacupuncture is a potentially cost-effective intervention, making it advantageous in resource-limited settings.

Despite electroacupuncture representing an accessible and practical treatment option that could improve the quality of life for advanced pancreatic cancer patients, further research is necessary to evaluate its long-term efficacy, as pain outcomes in the included studies were only assessed at up to two days post-treatment. The effectiveness of electroacupuncture for severe pancreatic-cancer-related pain remains unknown, as the aforementioned studies primarily enrolled patients with mild to moderate pain. Additional studies are also needed to investigate the underlying analgesic mechanisms of electroacupuncture.

### 4.5. Other Types of Unconventional Treatment

In addition to the previously discussed non-conventional therapies, other approaches, including non-pharmacological interventions such as spiritual practices, music and art therapies, and mind–body techniques, have been explored, although not specifically in patients with pancreatic cancer. Some of these modalities were investigated in the study by Axtner et al. [27], though their impact on overall survival and quality of life was not directly assessed. Future research evaluating the effectiveness of these interventions, alongside patient-reported outcomes (PROMs) and quality-of-life assessments, could provide valuable insights into the role of individual non-pharmacological interventions and multimodal therapeutic strategies in managing pancreatic cancer.

Alternative treatments such as scorpion venom [64,65] or multi-therapy regimens combining melatonin, bromocriptine, and vitamins [66] initially showed promise due to their cytotoxic effects against pancreatic cancer cells in vitro. However, subsequent pre-clinical studies and clinical trials have largely demonstrated their ineffectiveness and potential harm [66].

### 4.6. Efficacy and Harm of Unconventional Treatments for Pancreatic Cancer

This review highlights the widespread yet insufficiently regulated use of non-conventional therapies in pancreatic cancer treatment. The geographical distribution of the included studies reflects differences in medical culture, regulatory frameworks, and skepticism about the use of a specific treatment in different countries. This is reflected in the extensive use of mistletoe in Europe [23,24] and the use of other natural compounds, such as herbal medicine [9,11,12,13,14,15,20,21,22] and flavonoids such as curcumin [16,17,28,29], in contrast to Eastern countries.

Although preliminary evidence suggests potential benefits for unconventional therapies, the current guidelines do not endorse the use of these types of treatments as adjunct therapy in patients with pancreatic cancer, suggesting ongoing skepticism and resistance towards integrating conventional and unconventional therapies [67].

The primary barrier to the integration of non-conventional treatments lies in the frequent absence of robust safety data, which is often needed before the establishment of efficacy. It is essential to acknowledge that the use of natural substances has been a cornerstone of conventional medicine and pharmacology since their inception [68] and that this continues to play a significant role in therapeutic approaches today [69,70]. The critical distinction resides in the scientific rigor and ethical standards adhered to during the approval processes for substances within conventional medicine. These substances undergo a systematic and progressive series of experimental phases designed to determine therapeutic efficacy, establish appropriate dosing, evaluate safety profiles, and assess potential interactions with concurrent treatments and underlying conditions [71].

The Ethical Principles for Medical Research Involving Human Participants [72] have long prioritized the protection of patient health, rights, and well-being, with the safeguarding of life, dignity, and integrity as foundational pillars. As with modern pharmacological agents, exposure to potentially toxic herbal products can yield both therapeutic and harmful outcomes. The therapeutic or toxic response to any exogenous substance is governed by a multitude of factors, underscoring the necessity for comprehensive evaluation through rigorous research to ensure the safety and efficacy of these substances. A comprehensive understanding and formal acknowledgment of unconventional treatments for pancreatic cancer are imperative for healthcare stakeholders to provide patients with evidence-based information regarding the potential risks and benefits of these therapies. This is essential to prevent inadequate or unsafe use of such treatments. In addition to fostering clear and informed communication with patients, the advancement of integrative oncology is critical. This model of care should embrace a holistic approach, addressing not only the therapeutic needs of the patients but also their symptomatic, nutritional, psychological, and social requirements. Such an approach is crucial for ensuring the delivery of optimal, patient-centered care for pancreatic cancer.

The dearth of studies investigating less commonly adopted therapeutic modalities for pancreatic cancer highlights a substantial gap in both the existing literature and current clinical guidelines. This underscores the pressing need for further research to address the diverse, multifactorial needs of pancreatic cancer patients and to establish a robust evidence base that informs clinical practice relating to non-conventional therapies.

## 5. Conclusions

This review underscores the widespread adoption of non-conventional therapies in the management of pancreatic cancer, both as standalone therapies and in combination with standard treatments, despite the limited evidence supporting their clinical efficacy. Given the increasing prevalence of advanced pancreatic cancer, future research must rigorously evaluate the role of non-conventional treatments through well-structured clinical trials. Such investigations should comprehensively assess their safety, potential therapeutic benefits, and interactions with established treatment regimens as well as the underlying pathology. In the future, adopting a holistic approach to pancreatic cancer care will require providing evidence-based clinical practices that support the integration of verified non-conventional treatments into future clinical guidelines, ensuring comprehensive and patient-centered management.

## Figures and Tables

**Figure 1 cancers-17-01437-f001:**
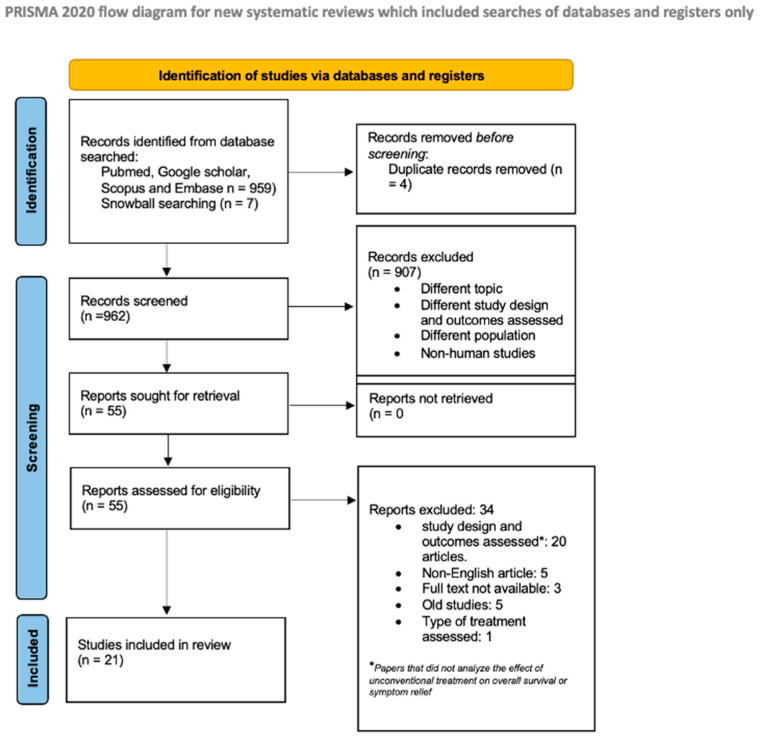
Flow diagram of the study selection process according to the PRISMA 2020 guidelines. Source: [7].

**Figure 2 cancers-17-01437-f002:**
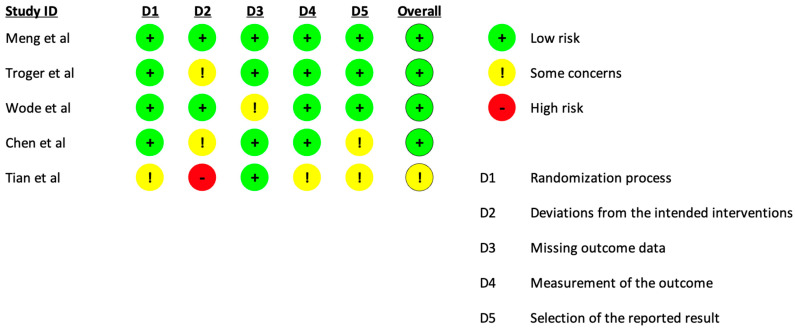
Risk of bias assessment for the five included RCTs, evaluated using the Cochrane Risk of Bias Assessment Tool 2 (Cochrane collaboration, 2019) for RCTs and tabulated using the ROBVIS tool. The assessment considered five domains: D1 (sequence generation), D2 (allocation concealment and blinding), D3 (incomplete outcome data), and D4 (selective outcome reporting). The risk of bias of each study was judged as low risk, high risk, or with some concerns [10,18,19,25,26].

**Figure 3 cancers-17-01437-f003:**
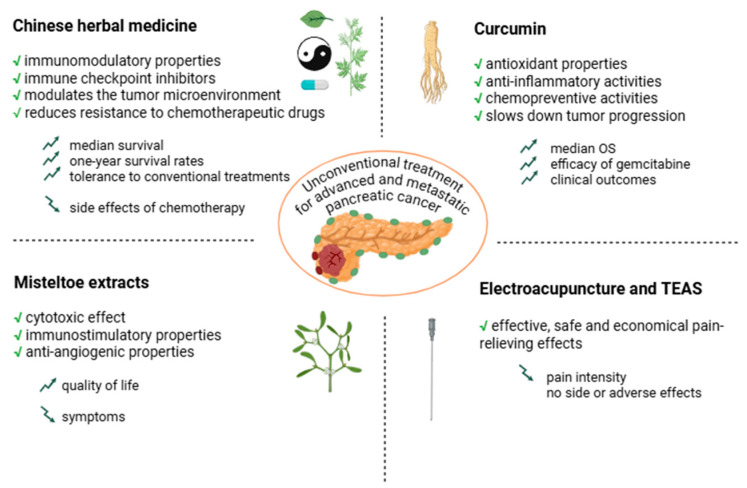
Summary of findings regarding the effects of unconventional treatments on overall survival and symptom relief.

**Table 1 cancers-17-01437-t001:** Characteristics of included studies. RCT: randomized controlled trial; NRSI: non-randomized controlled trial; CHM: Chinese herbal medicine; ME: mistletoe extract; WM: Western medicine; PDAC: pancreatic adenocarcinoma; PS: performance status; CT: chemotherapy; RT: radiotherapy; CNS: central nervous system.

Author	Year	Location	Study Design	Aim	Inclusion Criteria	Exclusion Criteria	Intervention Group	Control Group
Studies evaluating the effect of Chinese herbal medicine								
Wong et al. [9]	2018	China	NRSI	To evaluate the effects on overall survival and quality of life of CHM with/without conventional treatment in PDAC patients.	Patients diagnosed with PDAC.	Diagnosis of PDAC not coded.	CHM Concurrent treatment: WM.	Not applicable.
Saif et al. [20]	2013	USA	NRSI	To evaluate the efficacy of capecitabine and CHM PHY906 in patients with advanced PDAC who were previously treated with gemcitabine-based regimens.	Patients aged >18 years with histologic or cytologic diagnosis of locally advanced or metastatic PDAC who had failed prior chemotherapy; good PS; adequate bone marrow, kidney, and liver function.	Recent (<3 weeks) CT or RT; CNS metastases; active infection or uncontrolled concurrent medical illness; severe neurological impairment.	CHM (PHY906—a botanical formulation composed of four distinct herbs) 800 mg twice a day for four days in conjunction with capecitabine. Concurrent treatment: CT (capecitabine).	Not applicable.
Lee et al. [11]	2011	Korea	NRSI	To evaluate the effect of CHM Rhus verniciflua Stokes (RVS) extract combined with or without conventional therapy regimens in patients with advanced or metastatic PDAC.	Patients with non-resectable PDAC; good performance status; normal hepatic and renal function.	Prior CT or palliative radiation therapy; patients with neurologic deficit.	Daily oral administration of capsule containing RVS extract. Concurrent treatment: palliative CT.	Not applicable.
Meng et al. [10]	2012	China	RCT	To evaluate the feasibility and safety of treatment using huachansu in combination with gemcitabine in patients with advanced PDAC.	Patients aged >18 years with pathological diagnosis of unresectable PDAC; good performance status and adequate organ function.	Central nervous system metastases; other serious illnesses or conditions; psychiatric disorders; known allergies to huachansu or toad skin products.	Weekly gemcitabine 1000 mg intravenously over 30 min on days 1, 8, and 15 every 28 days combined with huachansu infusion intravenously, 20 mL over 2 h, intravenous infusion 5 days a week for 3 weeks, then 1 week off. Concurrent treatment: CT (gemcitabine).	Gemcitabine infusion and saline infusion with the same schedule of huachansu.
Kuo et al. [21]	2017	Taiwan	NRSI	To evaluate the effect of complementary CHM in combination with WM in patients with PDAC.	Patients diagnosed with PDAC with complete information and followed until the end of 2011.	Patients aged <20 years old; patients with a history of acute myocardial infarction.	Use of CHM in adjunction to WM: the most commonly used was Bai-hua-she-she-cao (Herba Oldenlandiae, Hedyotis diffusa Spreng). Concurrent treatment: WM.	No treatment with CHM.
Li et al. [22]	2018	China	NRSI	To evaluate the efficacy of CHM in patients with PDAC.	Patients aged 18 years or older; histological or radiological diagnosis of PDAC.	Patients with serious complications; concurrent cancer; severe mental disorder; incomplete medical records.	Integrative treatment group: patients received treatments of both individualized CHM and WM.	Not applicable.
Xue Yang et al. [12]	2015	China	NRSI	To evaluate the efficacy of CHM as an integrative treatment for selected patients with PDAC.	Patients aged 18 years or older; histological or radiological diagnosis of PDAC.	Patients with serious complications; concurrent cancer; pregnant or breastfeeding women; severe mental disorder; incomplete medical records.	CHM + WM. Concurrent treatment: WM.	WM (surgery, chemotherapy, radiotherapy).
Cao et al. [13]	2015	China	NRSI	To evaluate factors affecting overall survival in PDAC patients receiving a combined treatment of WM and CHM.	Adult patients with pathological diagnosis of PDAC; adequate PS; adequate hepatic, renal, and hematological function; estimated survival time of 3 months or more.	Patients aged <18 years; patients with neuroendocrine tumors or concurrent tumors; pregnant and lactating women; mentally ill patients; patients with severe acute and chronic diseases.	Combined CHM and WM. Concurrent treatment: WM.	WM-only group.
Ouyang et al. [14]	2011	China	NRSI	To evaluate the effect of a multimodal approach, including CHM, in patients with PDAC and liver metastasis.	Patients with PDAC and liver metastases.	Patients with neuroendocrine tumors or incomplete pathological reports.	CHM, in the form of a QYHJ decoction (composed of spreading Hedyotis herb, barbed skullcap herb, Ma-yuen Job’s tears seed, Lucid Ganoderma, and Chinese hawthorn fruit). Concurrent treatment: WM treatments.	No CHM.
Song et al. [15]	2017	China	NRSI	To evaluate the clinical effects of Babaodan Capsule (BBD) combined with Qingyi Huaji Formula (QYHJ) in treating patients with advanced PDAC.	Patients with histological or radiological diagnosis of PDAC; adequate PS; predicted survival time 3 months.	Patients with serious medical or psychiatric conditions, active infections, malnutrition, concurrent tumor, and life expectancy <3 months.	Qingyi Huaji Formula (QYHJ) plus Babaodan Capsule (BBD). Concurrent treatment: conventional treatment including CT, RT, radiofrequency ablation therapy, HIFU.	Qingyi Huaji Formula alone.
Studies evaluating the effect of mistletoe extract								
Schad et al. [23]	2013	Germany	NRSI	To evaluate the effect of intratumoral application of ME and monitor potential adverse drug reactions in patients with unresectable or metastatic PDAC.	Patients diagnosed with PDAC (stage II, III, IV).	Not reported.	Intratumoral application of ME (transabdominally or transgastrically/ transduodenally by endoscopic ultrasound-guided fine-needle application); some patients received subcutaneous and intravenous application. Concurrent treatment: CT.	Not applicable.
Thronicke et al. [24]	2020	Germany	NRSI	To evaluate the cost-effectiveness of ME in addition to standard of care compared to standard of care alone in stage IV PDAC patients from the hospital’s perspective.	Adult patients with stage IV PDAC receiving standard of care and surviving more than 21 days.	Patients whose death date or the last contact date was not available.	Subcutaneous application of ME (intravenous and intratumoral application was performed in individual cases). Concurrent treatment: CT.	Patients received only standard of care and no add-on mistletoe therapy.
Troger et al. [25]	2014	Serbia	RCT	To evaluate the efficacy of ME monotherapy on the survival and quality of life (QoL) of patients with locally advanced or metastatic PDAC.	Adult patients with stage III or IV PDAC; unsuitability for, or unwillingness to undergo, any other type of cancer treatment; adequate bone marrow function.	Life expectancy <4 weeks; weight loss of ≥20% in the past 6 weeks; brain metastases.	Subcutaneous application of ME. Concurrent treatment: best supportive care (symptomatic treatment of pain, nausea, vomiting, and dyspepsia).	No treatment.
Wode et al. [26]	2024	Sweden	RCT	To evaluate the effect of ME as a complementary therapy to standard treatment (palliative CT or best supportive care) on overall survival and quality of life in patients with advanced pancreatic cancer.	Adult patients with stage III or IV PDAC or relapse of pancreatic cancer; adequate PS.	Life expectancy less than 4 weeks; pregnancy or breastfeeding; neuroendocrine tumors of the pancreas; brain metastases; medical, psychiatric, or cognitive disorders.	Subcutaneous application of ME. Best supportive care for symptom management and palliative care.	Placebo.
Axtner et al. [27]	2016	Germany	NRSI	To evaluate the effect of integrative oncology (including the use of ME treatment and non-pharmacological interventions) on overall survival in patients with stage IV PDAC.	Patients with stage IV PDAC.	Patients who lived less than 4 weeks.	ME (subcutaneously or intravenously or intratumor application) and T for a minimum of 4 weeks. Concurrent treatment: T.	Conventional treatment alone or with ME for less than 1 week.
							Mistletoe therapy (subcutaneously or intravenously or intratumor application) alone for more than 4 weeks. Concurrent treatment: No concurrent treatment.	No treatment or mistletoe for less than 1 week.
Studies evaluating the effect of curcumin								
Pastorelli et al. [28]	2018	Italy	NRSI	To evaluate the safety and activity of curcumin as a nutritional complement to gemcitabine in patients with locally advanced or metastatic PDAC.	Adult patients diagnosed with locally advanced or metastatic PDAC; previous adjuvant treatment completed at least 6 months prior to data collection; adequate PS; adequate hepatic, renal, and bone marrow function.	Concurrent malignancies; brain metastases; intercurrent significant systemic illness.	Curcumin (Meriva) 2000 mg/day orally administered every day. Concurrent treatment: gemcitabine.	No comparison.
Kanai et al. [16]	2010	Japan	NRSI	To evaluate the safety and feasibility of oral curcumin in combination with gemcitabine chemotherapy in patients with PDAC.	Adult patients diagnosed with PDAC who showed disease progression during gemcitabine-based T and had no other effective treatment option; adequate performance status; adequate bone marrow, liver, and renal function.	History of severe drug allergy; pregnancy or lactation; other severe comorbid diseases.	Oral curcumin daily at a dose of 8 g. Concurrent treatment: gemcitabine.	Placebo.
Kanai et al. [17]	2013	Japan	NRSI	To evaluate the safety of repetitive exposure to high concentrations of curcumin achieved by Theracurmin.	Adult patients diagnosed with PDAC who showed disease progression after CT and had no other effective treatment option; adequate performance status; adequate bone marrow, liver, and renal function.	History of severe drug allergy; pregnancy or lactation; other severe comorbid diseases.	Oral administration of Theracurmin in water solution at three different dosages (containing, respectively, 100, 200, and 400 mg of curcumin). Concurrent treatment: gemcitabine.	Not applicable.
Epelbaum et al. [29]	2010	Israel	NRSI	To evaluate the activity and feasibility of gemcitabine in combination with curcumin in patients with advanced PDAC.	Adult patients diagnosed with locally advanced or metastatic PDAC; adequate PS; adequate hepatic, renal, and bone marrow function.	Patients with an unstable medical condition or intercurrent illness were excluded.	Oral curcumin daily with a dose of 8 g. Concurrent treatment: gemcitabine.	No comparison.
Studies evaluating unconventional therapies for pancreatic cancer pain								
Chen et al. [18]	2013	China	RCT	To evaluate the analgesic effect of electroacupuncture on pancreatic cancer pain.	Adult patients diagnosed with advanced PDAC with pain intensity 3–6 on a numeric rating scale (NRS); patients who received a stable dose of analgesics at least 72 h before randomization.	Presence of disease leading to pain; contraindications for the use of acupuncture or history of cerebrovascular accident or spinal cord injury.	Electroacupuncture. Concurrent treatments: analgesic drugs.	Placebo.
Tian et al. [19]	2024	China	RCT	To evaluate the analgesic effect of transcutaneous electrical acupoint stimulation in patients with advanced pancreatic cancer.	Adult patients diagnosed with advanced PDAC with pain intensity 3–8 on a numeric rating scale (NRS); intact skin at the connecting electrode sites; without damage and sensory abnormalities; no history of mental illness or drug abuse.	Presence of disease leading to pain; contraindications to percutaneous electrical stimulation or history of cerebrovascular accident or spinal cord injury; patients with implantation of a cardiac pacemaker.	Electroacupuncture. Concurrent treatments: analgesic drugs.	Routine pain medication after admission.

**Table 2 cancers-17-01437-t002:** Demographic data of patient cohorts in included studies. NA: not available. Age is expressed in years as mean (standard deviation) or median (interquartile range); gender is expressed as prevalence of male patients (percentage).

Author	Age (Years)		Gender Male (%)		Sample Size				
	Control Group	Intervention Group	Control Group	Intervention Group	Total	Control Group	Intervention Group	Withdrawal Control Group	Withdrawal Intervention Group
Studies evaluating the effect of Chinese herbal medicine									
Wong et al. [9]		36			21	0	21	NA	NA
Saif et al. [20]	-	64 (45–84)	-	15 (60%)	25	0	25	NA	NA
Lee et al. [11]		63.0 (36–78)	0	16 (38.1%)	42	0	42	NA	NA
Meng et al. [10]	61.7(9.9)	60.2 (9.5)	23 (62.1%)	23 (58.9%)	76	37	39	10	8
Kuo et al. [21]	63.60 (12.03)	62.79 (10.98)	218 (56.48%)	218 (56.48%)	772	386	386	NA	NA
Li et al. [22]	NA	NA	NA	NA	174	43	131	NA	NA
Xue Yang et al. [12]		62			107	56	51	NA	11
Cao et al. [13]	NA	NA	NA	NA	272	136	136	NA	NA
					130	65	65		
					142	71	71		
Ouyang et al. [14]	NA	NA	46 (70.8%)	49 (75.4%)	164	42	122	NA	NA
Song et al. [15]	60.05 (41–77)	59.63 (41–82)	45 (63.4%)	45 (63.4%)	81	41	40	NA	NA
Studies evaluating the effect of mistletoe extract									
Schad et al. [23]	-	61 (39–85)	-	17 (44%)	39	0	39	NA	NA
Thronicke et al. [24]	68.61 (10.06)	63.71 (11.87)			88	34	54	NA	NA
Troger et al. [25]	NA	NA	63	65	220	110	110	3	0
Wode et al. [26]	68	70	74	70	290	147	143	NA	NA
Axtner et al. [27]	65 (39–77)	66 (34–89)	14 (35%)	61 (57%)	147	107	40	NA	NA
	74 (39–85)	73 (39–89)	10 (41%)	7 (16%)	67	43	24		
Studies evaluating the effect of curcumin									
Pastorelli et al. [28]	-	66 (42–87)	-	29 (66%)	44	0	44	NA	NA
Kanai et al. [16]	-	67 (44–79)	-	13 (62%)	21	0	21	NA	NA
Kanai et al. [17]	-	64 (50–84)	-	11 (68%)	16	0	16	NA	NA
Epelbaum et al. [29]	-	69 (54–78)	-	10 (59%)	17	0	17	NA	NA
Studies evaluating unconventional therapies for pancreatic cancer pain									
Chen et al. [18]	59.1 (9.1)	60.1 (8.5)	20 (66.7)	19 (63.3)	60	29	30	1	0
Tian et al. [19]	62.8 ± 10.6	59.3 ± 10.4	25 (62.5%)	26 (65.0%)	80	40	40	NA	NA

**Table 3 cancers-17-01437-t003:** Outcomes measures in included studies. NA: not available; RR: risk ratio; IC: confidence interval.

Author	Overall Survival					Symptom Relief					Quality of Life				
	Control Group (Months)	Intervention Group (Months)	*RR*	*IC*	*p* Value	Control Group	Intervention Group	*RR*	*IC*	*p* Value	Control Group	Intervention Group	*RR*	*IC*	*p* Value
Studies evaluating the effect of Chinese herbal medicine															
Wong et al. [9]	NA	15.2	NA	NA	NA	NA	NA	NA	NA	NA	NA	NA	NA	NA	NA
Saif et al. [20]	NA	5.4	NA	NA	NA	NA	NA	NA	NA	NA	NA	NA	NA	NA	NA
Lee et al. [11]	NA	7.8	6.55	2.65–16.23	0.01	NA	NA	NA	NA	NA	NA	NA	NA	NA	NA
Meng et al. [10]	13	13.3	NA	NA	0.01	NA	NA	NA	NA	NA	43.33 (28.39)	43.33 (28.39)	NA	NA	NA
Kuo et al. [21]	NA	NA	NA	NA	0.01	NA	NA	NA	NA	NA	NA	NA	NA	NA	NA
Li et al. [22]	8.03	18.3	NA	NA		NA	NA	NA	NA	NA	NA	NA	NA	NA	NA
Xue Yang et al. [12]	8	19			<0.001	NA	NA	NA	NA	NA	NA	NA	NA	NA	NA
Cao et al. [13]	17.4	11.3	0.45	0.34–0.60	NA	NA	NA	NA	NA	NA	NA	NA	NA	NA	NA
	9.9	12.7	0.52	0.35–0.76											
	12.4	23.8	0.37	0.25–0.55											
Ouyang et al. [14]	3.9	5.4	−0.66	[−1.13–−0.202]	NA	NA	NA	NA	NA	NA	NA	NA	NA	NA	NA
Song et al. [15]	6.9	19.27	NA	NA	NA	NA	NA	NA	NA	NA	NA	NA	NA	NA	NA
Studies evaluating the effect of mistletoe extract															
Schad et al. [23]	NA	11	NA	NA	NA	NA	NA	NA	NA	NA	NA	NA	NA	NA	NA
Thronicke et al. [24]	6.03	8.43	NA	NA	NA	NA	NA	NA	NA	NA	NA	NA	NA	NA	NA
Troger et al. [25]	2.7	4.8	0.49	NA	0.01	NA	NA	NA	NA	NA	NA	NA	NA	NA	NA
Wode et al. [26]	8.3	7.8	NA	NA	NA	NA	NA	NA	NA	NA	NA	NA	NA	NA	NA
Axtner et al. [27]	7.3	12.1	NA	NA	NA	NA	NA	NA	NA	NA	NA	NA	NA	NA	NA
	2.5	5.4													
Studies evaluating the effect of curcumin															
Pastorelli et al. [28]	NA	10.2	NA	NA	NA	NA	NA	NA	NA	NA	NA	NA	NA	NA	NA
Kanai et al. [16]	NA	13.4	NA	NA	NA	NA	NA	NA	NA	NA	NA	24.8 ± 14.3	NA	NA	NA
Kanai et al. [17]	4.4	NA	NA	NA	NA	NA	NA	NA	NA	NA	NA	NA	NA	NA	NA
Epelbaum et al. [29]	NA	5	NA	NA	NA	NA	NA	NA	NA	NA	NA	NA	NA	NA	NA
Studies evaluating unconventional therapies for pancreatic cancer pain															
Chen et al. [18]	NA	NA	NA	NA	NA	NA	NA	NA	NA	NA	NA	NA	NA	NA	NA
Tian et al. [19]	NA	NA	NA	NA	NA	NA	NA	NA	NA	NA	NA	NA	NA	NA	NA

## Data Availability

The datasets analyzed during the current study are not publicly available but are available from the corresponding author on reasonable request.

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
