# Peer review of "Unconventional Treatments for Pancreatic Cancer: A Systematic Review"

_cancers, 2025, doi:10.3390/cancers17091437_

Round 1
Reviewer 1 Report
Comments and Suggestions for Authors
The study by Vella et al discusses about the unconventional treatment options for pancreatic cancer.
Some suggestions to improve this manuscript are:
- Please elaborate on the conclusions section by adding more on future directions.
- More explanation is required for Figure 1.
Author Response
We would like to sincerely thank the reviewers for their thoughtful comments, constructive feedback, and valuable suggestions on our manuscript. We greatly appreciate the time and effort invested in reviewing our work. In response to the reviewers’ comments, we have revised the manuscript accordingly. Below, we provide a detailed, point-by-point response outlining the changes made. All modifications in the manuscript have been highlighted for clarity.
Suggestion 1: Please elaborate on the conclusions section by adding more on future directions.
We have added in the conclusion section a more explicit statement on the auspicable direction of future research to rigorously evaluate the safety and efficacy of unconventional treatment in a holistic approach to pancreatic cancer patients : " Given the increasing prevalence of advanced pancreatic cancer, future research must rigorously evaluate the role of non-conventional treatments through well-structured clinical trials. Such investigations should comprehensively assess their safety, potential therapeutic benefits, and interactions with established treatment regimens and the un-derlying pathology. In the future, adopting a holistic approach to pancreatic cancer care will require providing evidence-based clinical practices that support the integration of verified non-conventional treatments into future clinical guidelines, ensuring compre-hensive and patient-centered management. "
Suggestion 2: More explanation is required for Figure 1.
We have added figure description for figure 1 , figure 2 and figure 3
Reviewer 2 Report
Comments and Suggestions for Authors
The
Vella et al. discuss in their review article the use of unconventional therapeutic approaches for the treatment of pancreatic cancer, through a systemic analysis of studies from 2010 to March 2024. The approaches examined include Chinese herbal medicine (CHM), curcumin, mistletoe extract, and electroacupuncture. According to the inclusion criteria set by the authors for the study, 21 studies involving 3,095 patients were analyzed. The results indicated that CHM and curcumin, when combined with conventional therapy, improved patient survival and quality of life. Electroacupuncture was particularly effective in pain reduction, while mistletoe extract showed benefits for patients in only some studies.
The work presented by the authors highlights a previously underexplored branch of therapy options for pancreatic cancer. The study design is well-reasoned, and the inclusion criteria appear suitable for a comprehensive analysis of the topic.
Points:
- Line 132: A reference is made to Figure 1. The figure follows immediately after in the text; however, it lacks the figure legend and a clear indication that the displayed scheme represents Figure 1. This should be corrected. The same applies to Figure 2 and 3.
- Table 3: The first row, which typically includes the table title and other methodological information (as seen in Tables 1 and 2), is missing. This should be added.
- In the tables, some words break unexpectedly in the middle of lines without a dash being used. This should be corrected. Additionally, in Table 3, abbreviations like RR and IC should be explained. The formatting of Table 3 should ensure that "NA" always appears on a single line.
- Line 132: The reference is made to Appendix 2. However, Appendix B is mentioned in line 459, which seems to be the intended reference.
- Line 147: The "w" should be removed.
- Lines 263 and 282: The abbreviations CAM, CHM, and ME were introduced earlier in the text. These abbreviations should be used consistently from their first mention onward.
Author Response
We would like to sincerely thank the reviewers for their thoughtful comments, constructive feedback, and valuable suggestions on our manuscript. We greatly appreciate the time and effort invested in reviewing our work. In response to the reviewers’ comments, we have revised the manuscript accordingly. Below, we provide a detailed, point-by-point response outlining the changes made. All modifications in the manuscript have been highlighted for clarity.
- "Line 132: A reference is made to Figure 1. The figure follows immediately after in the text; however, it lacks the figure legend and a clear indication that the displayed scheme represents Figure 1. This should be corrected. The same applies to Figure 2 and 3. " We have added legend for figure 1, 2 and 3.
- "Table 3: The first row, which typically includes the table title and other methodological information (as seen in Tables 1 and 2), is missing. This should be added. In the tables, some words break unexpectedly in the middle of lines without a dash being used. This should be corrected. Additionally, in Table 3, abbreviations like RR and IC should be explained. The formatting of Table 3 should ensure that "NA" always appears on a single line."
We thank the reviewer for their valuable observations regarding Table 3.
We have added a proper header row to Table 3, in line with the formatting used in Tables 1 and 2, including the table title and relevant methodological details.
The word-breaking issues within the table have been corrected to prevent unexpected line breaks, and appropriate hyphenation has been applied where necessary.
The abbreviations "RR" (Relative Risk) and "IC" (Confidence Interval) have now been clearly defined in the table footnote for clarity.
We have also adjusted the formatting to ensure that all instances of "NA" now appear consistently on a single line.
- "Line 132: The reference is made to Appendix 2. However, Appendix B is mentioned in line 459, which seems to be the intended reference."
We appreciate the reviewer’s careful reading and helpful observation. The reference on line 132 has been corrected to "Appendix B" to ensure consistency with the corresponding reference on line 459. Thank you for pointing out this discrepancy.
- "Lines 263 and 282: The abbreviations CAM, CHM, and ME were introduced earlier in the text. These abbreviations should be used consistently from their first mention onward."
Thank you for highlighting this inconsistency. We have revised the manuscript to ensure that the abbreviations CAM (Complementary and Alternative Medicine), CHM (Chinese Herbal Medicine), and ME (Mistletoe extract) are used consistently throughout the text from their first introduction onward.
- "Line 147: The "w" should be removed."
Thank you for pointing this out. The extraneous "w" on line 147 has been removed as suggested.
Reviewer 3 Report
Comments and Suggestions for Authors This paper reviewes the clinical trial data investigating the complementary medicine including Chinese herbal medicine, curcumin, and mistletoe extract. The topic is interesting; however, the following issues need to be addressed.- In general, tables contain too many columns to read. Please select the relevant information and improve the readability. Rotating the table may help.
- Since most of the cells are filled with "NA", Table 3 could be omitted.
- This reviewer suggests that summarizing the study design and primary outcome in one table.
- Some infomation is inaccurate. For example, references 16 and 17 are from Japan, not China (Table 1). In addition, ref 17 did not test curcumin but was cited in the following sentence "The recommended daily dose of curcumin (8 g/day) was used in all studies" (p.18, line 220-221). Please check the accuracy of the information throughout the manuscript.
- In the discussion section, Chinese herbal medicine, mistletoe extract, and curcumin are introduced for the first time. However, placing the introduction of these agents in the earlier section (e.g., Results) may be helpful for readers.
- There are no titles for Figure 1 and 2.
- Figure 1 contains some wavy red lines. Please remove them.
- Please add the reference number to the author's name in Figure 2.
- Some texts seem to contain grammatical errors (e.g., p.15, line 152). Please check throughout the manuscript.
Author Response
We would like to sincerely thank the reviewers for their thoughtful comments, constructive feedback, and valuable suggestions on our manuscript. We greatly appreciate the time and effort invested in reviewing our work. In response to the reviewers’ comments, we have revised the manuscript accordingly. Below, we provide a detailed, point-by-point response outlining the changes made. All modifications in the manuscript have been highlighted for clarity.
- Comments 1-3: "In general, tables contain too many columns to read. Please select the relevant information and improve the readability. Rotating the table may help.Since most of the cells are filled with "NA", Table 3 could be omitted.This reviewer suggests that summarizing the study design and primary outcome in one table."
We appreciate the reviewer’s insightful feedback regarding the tables. In response, we have revised the tables to improve readability by reducing the number of columns and avoiding excessive use of "NA" entries.
- Comment 4 "some infomation is inaccurate. For example, references 16 and 17 are from Japan, not China (Table 1). In addition, ref 17 did not test curcumin but was cited in the following sentence "The recommended daily dose of curcumin (8 g/day) was used in all studies" (p.18, line 220-221). Please check the accuracy of the information throughout the manuscript."
We thank the reviewer for pointing out these important details. We have corrected the country of origin for references 16 and 17 in Table 1, accurately identifying them as studies from Japan rather than China. Regarding reference 17 ("A phase I/II study of gemcitabine-based chemotherapy plus curcumin for patients with gemcitabine-resistant pancreatic cancer"), we would like to clarify that this study does in fact refer to curcumin treatment, as it investigates the combination of curcumin with gemcitabine. Nonetheless, we have reviewed the manuscript thoroughly to ensure the accuracy of all referenced information.
- Comment 5 : "In the discussion section, Chinese herbal medicine, mistletoe extract, and curcumin are introduced for the first time. However, placing the introduction of these agents in the earlier section (e.g., Results) may be helpful for readers."
We appreciate the reviewer’s thoughtful suggestion. However, we chose to keep the Results section focused strictly on the presentation of study findings, without additional contextual information, in order to maintain clarity and structure. The introduction of Chinese herbal medicine, mistletoe extract, and curcumin in the Discussion section was intended to provide interpretation and context for the findings rather than introduce new information. We hope this approach supports a clear separation between results and their interpretation.
- Comments 6-8: " There are no titles for Figure 1 and 2.Figure 1 contains some wavy red lines. Please remove them.Please add the reference number to the author's name in Figure 2."
Thank you for your helpful observations. We have added appropriate titles to Figures 1 and 2 to enhance clarity. The wavy red lines in Figure 1 have been removed. Additionally, reference numbers have been added next to the authors’ names in Figure 2, as suggested.
- Comment 9: " Some texts seem to contain grammatical errors (e.g., p.15, line 152). Please check throughout the manuscript."
We appreciate the reviewer’s comment regarding the language quality. To address this, we have submitted the manuscript for extended English language revision to ensure grammatical accuracy and improve overall readability throughout the text.
Round 2
Reviewer 3 Report
Comments and Suggestions for Authors
The following point should be revised before publication.
Regarding reference 17 ("A phase I/II study of gemcitabine-based chemotherapy plus curcumin for patients with gemcitabine-resistant pancreatic cancer"), we would like to clarify that this study does in fact refer to curcumin treatment, as it investigates the combination of curcumin with gemcitabine.
According to the submitted manuscript, reference 17 is "A phase I study investigating the safety and pharmacokinetics of highly bioavailable curcumin (Theracurmin) in cancer patients. " and should be revised accordingly.
Author Response
We thank the reviewer for the constructive feedback and valuable suggestions on our manuscript. In response, we have clarified in the "Curcumin" section that the included studies evaluated both curcumin and its derivatives. Additionally, we specified in the following sentence that one study—referenced as number 17—specifically evaluated Theracurmin, a curcumin derivative. We hope the reviewer finds this revision satisfactory and that the clarity of the paragraph and references has been maintained.
"Curcumin
Four single-cohort studies [16, 17, 28, 29] investigated curcumin or curcumin derivatives as a complementary therapy in combination with gemcitabine for pancreatic cancer. One study [17] examined Theracurmin, a highly bioavailable curcumin derivative, for its safety and pharmacokinetics"